# Nonalcoholic Fatty Liver Disease and the Kidney: A Review

**DOI:** 10.3390/biomedicines9101370

**Published:** 2021-10-01

**Authors:** Ilaria Umbro, Francesco Baratta, Francesco Angelico, Maria Del Ben

**Affiliations:** 1Geramed Dialysis Center, Fiano Romano, 00065 Rome, Italy; 2Department of Clinical Internal, Anesthesiological and Cardiovascular Sciences, Sapienza University of Rome, 00161 Rome, Italy; francesco.baratta@uniroma1.it (F.B.); maria.delben@uniroma1.it (M.D.B.); 3Department of Public Health and Infectious Diseases, Sapienza University of Rome, 00161 Rome, Italy; francesco.angelico@uniroma1.it

**Keywords:** nonalcoholic fatty liver disease, liver fibrosis, chronic kidney disease, renal function, kidney

## Abstract

Nonalcoholic fatty liver disease (NAFLD) is associated with several extrahepatic manifestations such as cardiovascular disease and sleep apnea. Furthermore, NAFLD is reported to be associated with an increased risk of incident chronic kidney disease (CKD). Inflammation and oxidative stress are suggested to be the key factors involved in the inflammatory mechanisms and pathways linking NAFLD to CKD and are responsible for both the pathogenesis and the progression of CKD in NAFLD patients. This review aims to provide a more comprehensive overview of the association between CKD and NAFLD, also considering the effect of increasing severity of NAFLD. A PubMed search was conducted using the terms “non-alcoholic fatty liver disease AND kidney”. In total, 537 articles were retrieved in the last five years and 12 articles were included in the qualitative analysis. Our results showed that CKD developed more frequently in NAFLD patients compared to those without NAFLD. This association persisted after adjustment for traditional risk factors and according to the severity of NAFLD. Therefore, patients with NAFLD should be considered at high risk of CKD. Intensive multidisciplinary surveillance over time is needed, where hepatologists and nephrologists must act together for better and earlier treatment of NAFLD patients.

## 1. Introduction

The liver and kidneys are recognized to be closely and mutually intertwined, both in physiological and pathological conditions [1]. Renal dysfunction in the context of liver disease is multifactorial. It can be related to primary diseases such as polycystic liver–kidney disease [2] or can occur as a consequence of hepatitis B and C virus infections [3,4] or excessive alcohol consumption [5]. More recently, nonalcoholic fatty liver disease (NAFLD) was demonstrated to be significantly associated with increased risk of incident chronic kidney disease (CKD) [6].

### 1.1. Nonalcoholic Fatty Liver Disease

NAFLD represents the most common chronic liver disease worldwide, affecting approximately 25% of global adult population. It should no longer be considered only a Western disease, and its prevalence is expected to further increase over the years [7,8]. NAFLD is characterized by hepatic steatosis in more than 5% of hepatocytes in the absence of excessive alcohol consumption after excluding other competing causes of chronic liver disease such as viral or autoimmune hepatitis [9].

In general, specific signs or clinical symptoms of NAFLD are absent in most patients at the time of diagnosis. Liver ultrasonography represents the first-line diagnostic procedure, whereas liver biopsy is the gold-standard procedure for NAFLD staging. Furthermore, several noninvasive hepatic steatosis and/or fibrosis tests, such as the fibrosis-4 (FIB-4) score, the NAFLD fibrosis score (NFS), the enhanced liver fibrosis (ELF) score, the fatty liver index (FLI) or the transient elastography (FibroScan), are used to identify patients who require liver biopsy [10].

The risk of developing NAFLD is increased in case of metabolic syndrome and its components such as hypertension, obesity, type 2 diabetes mellitus and dyslipidemia [11,12]. NAFLD is recognized to be associated with several extrahepatic manifestations such as cardiovascular disease [13,14] and sleep apnea [15]. Recently, the renaming of NAFLD to metabolic associated fatty liver disease (MAFLD) has been proposed to stress the strong association with overweight/obesity, type 2 diabetes and the metabolic syndrome [16,17,18]. Furthermore, NAFLD is reported to be associated with an increased incidence of CKD. However, it has not yet been demonstrated if the strength of this association could be affected by increasing severity of NAFLD [6].

The pathogenesis of NAFLD is multifactorial. Unhealthy lifestyles, dietary habits, metabolic dysfunction, oxidative stress, genetic factors and alterations of gut–liver axis are the main pathophysiological mechanisms. Insulin resistance, which causes liver steatosis, is considered to play a central role. Fat accumulation is reported to sensitize liver to induce inflammation and oxidative stress, leading to the progression of NAFLD and liver fibrosis [19]. Moreover, several lines of evidence indicated that also genetic factors may predispose to NAFLD, and among the others, a variant located at the PNPLA3 gene appears to show the strongest effect [20,21]. Finally, dysbiosis, increased intestinal barrier permeability, bacterial translocation and endotoxemia may also promote Kupfer cell activation and inflammatory cytokine production [22,23].

Inflammation and oxidative stress are suggested to be the key factors involved in the inflammatory mechanisms and pathways linking NAFLD to CKD and responsible for both the pathogenesis and the progression of CKD in patients with NAFLD.

### 1.2. Chronic Kidney Disease

CKD was listed as one of the 10 leading causes of reduced life expectancy or disability-adjusted life-years in 2013 and was ranked as the 12th cause of mortality in 2017 [24,25].

The burden of CKD varies significantly across the world but, in 2017, its prevalence was estimated to be 9.1% worldwide [26].

According to the “Kidney Disease: Improving Global Outcomes 2012 Clinical Practice Guideline”, CKD was defined as the reduction of glomerular filtration rate (GFR) < 60 mL/min/1.73 m^2^ or the evidence of kidney damage (structural or functional), present for > 3 months, with implications for health [27].

CKD can be classified on the basis of GFR categories as follows: G1, GFR ≥ 90 mL/min/1.73 m^2^; G2, GFR = 60–89 mL/min/1.73 m^2^; G3a, GFR = 45–59 mL/min/1.73 m^2^; G3b, GFR = 30–44 mL/min/1.73 m^2^; G4, GFR = 15–29 mL/min/1.73 m^2^; G5, GFR < 15 mL/min/1.73 m^2^. Furthermore, CKD can be classified according to albuminuria categories as follows: A1, albumin-to-creatinine ratio (ACR) < 30 mg/g; A2, ACR = 30–300 mg/g; A3, ACR > 300 mg/g [27].

In general, the metabolic risk factors for NAFLD may influence the development of CKD, and several studies suggested that the presence of NAFLD may accelerate the development and progression of CKD independently of the traditional cardiometabolic risk factors [28,29]. However, there is a heterogeneity of literature data related to different study designs, definitions of CKD and diagnosis of NAFLD. 

This review aims to provide a more comprehensive overview of the association between CKD and NAFLD, considering also the effect of increasing severity of NAFLD.

## 2. Materials and Methods

### Literature Search, Data Selection and Extraction

A PubMed search was conducted using the terms “non-alcoholic fatty liver disease AND kidney”. 

Research was limited to studies published in the English language over the last 5 years. Studies regarding animals and children, case reports, commentaries, letters, editorials and meeting abstracts were not considered. Review-type articles were examined only to identify papers potentially eligible for inclusion.

Only relevant studies potentially eligible for inclusion underwent full-text review. The following data were extracted on a data collection table: first author/publication year, study design, number of patients enrolled, diagnosis of NAFLD including assessment of severity of liver fibrosis, diagnosis of CKD, patient characteristics, main results of the study.

## 3. Results

### 3.1. Study Flow

A flow chart of the search for relevant studies is presented in Figure 1. Overall, 537 articles were found on PubMed search in the last 5 years. After exclusion of nonsignificant studies according to title, language and abstract (*n* = 520), 17 articles were selected for full-text review. A further five studies were excluded for the following reasons: one did not have a clear explanation of diagnosis of NAFLD and/or CKD, one had an inaccurate diagnosis of CKD and three did not clearly describe patient enrollment and assessment. Finally, 12 papers were included in the qualitative analysis.

### 3.2. Study Characteristics

A summary of the 12 relevant studies is reported in Table 1. There was a total of 180,010 participants across all the studies with a proportion of males ranging from 35 to 79%. 

NAFLD was diagnosed by abdominal ultrasound in five studies [30,31,32,33,34], only three studies used the gold-standard liver biopsy [35,36,37], two studies used the fatty liver index (FLI) [38,39], one study used the fibrosis-4 (FIB-4) score [40] and one study enrolled NAFLD patients by the 10th revision of the International Classification of Diseases (ICD-10) [41]. The diagnosis of CKD was performed according to The Kidney Disease: Improving Global Outcomes 2012 Clinical Practice Guideline as the reduction of GFR < 60 mL/min/1.73 m^2^ estimated by different formulae in almost all the studies (KDIGO 2012), except for one study that used the International Classification of Diseases, 10th revision (ICD-10) [41].

The exclusion criteria considered in the relevant studies were as follows: patients with other chronic liver disease besides NAFLD, patients who were infected with hepatitis B and/or C virus, patients with excessive alcohol consumption, patients with current use of hepatotoxic drugs and patients with previous diagnosis of CKD. Furthermore, we excluded studies with a follow-up < 1 year and studies not reporting any regression analysis for the outcome of interest.

**Table 1 biomedicines-09-01370-t001:** Summary of the 12 relevant studies considered in the review.

Author/Year (Country)	Study Design	Number of Patients	Diagnosis of NAFLD	Diagnosis of CKD	Patient Characteristics	Main Results of the Study
Huh JH et al., 2017 (Korea) [38]	Prospective cohort study	4761 patients (1808 men and 2953 women)	FLI ≥ 60	GFR < 60 mL/min/1.73 m^2^ estimated by CKD-EPI equation	Prevalence of NAFLD was 12.6% (601 patients). 121/601 subjects (20.1%) developed CKD.	NAFLD is associated with CKD independently of traditional risk factors (HR 1.46, 95% CI 1.19–1.79).
Sinn DH et al., 2017 (Korea) [30]	Retrospective longitudinal cohort study	41,430 patients (25,217 men and 16,213 women)	Ultrasound NFS, APRI and FIB-4 score to assess severity of fibrosis	GFR < 60 mL/min/1.73 m^2^ estimated by CKD-EPI equation	Prevalence of NAFLD was 34.3% (14,223 patients). 691 subjects developed incident CKD.	NAFLD was independently associated with incident CKD (aHR 1.22, 95% CI 1.04–1.43). This association was progressive with increased severity of liver disease (aHR 1.59, 95% CI 1.31–1.93).
Wijarnpreecha K et al., 2018 (USA) [31]	Cross-sectional study	4142 NAFLD patients (1932 men and 2210 women)	Ultrasound NFS, FIB-4 score, APRI and BARD score to assess severity of fibrosis	GFR < 60 mL/min/1.73 m^2^ estimated by MDRD equation	Prevalence of NAFLD was 100%. 200/4142 subjects (4.8%) developed CKD.	Advanced liver fibrosis, defined by NFS and FIB-4 scores, is associated independently with CKD among individuals with NAFLD. FIB-4 (aHR 2.27, 95% CI 1.05–4.52); NFS (aHR 4.92, 95%CI 2.96–8.15). FIB-4 is the best predictor of an increased risk of prevalent CKD.
Ӧnnerhag K et al., 2019 (Sweden) [35]	Retrospective cohort study	144 NAFLD patients (83 men and 61 women)	Biopsy FIB-4 score, NFS, APRI and BARD score to assess severity of fibrosis	GFR < 60 mL/min/1.73 m^2^ estimated by CKD-EPI equation	Prevalence of NAFLD was 100%. 47/144 subjects (32.6%) developed CKD.	Intermediate (1.30–2.67) and high (>2.67) FIB-4 scores were independently associated with CKD (aHR 4.77, 95% CI 1.95–11.64; aHR 7.25, 95% CI 2.51–20.94, respectively). Similar results with intermediate and high NFS (aHR 3.31, 95% CI 1.41–7.74; aHR 31.38, 95% CI 7.92–124.38, respectively).
Choi JW et al., 2019 (Korea) [40]	Cross-sectional study	11,836 NAFLD patients (4893 men and 6943 women)	FIB-4 score	GFR < 60 mL/min/1.73 m^2^ estimated by CKD-EPI equation	Prevalence of NAFLD was 100%.	FIB-4 was an independent predictor of incipient CKD (aOR 1.254, 95% CI 1.034–1.521).
Sun DQ et al., 2019 (China) [36]	Cross-sectional study	217 NAFLD patients (171 men and 46 women)	Biopsy	GFR < 60 mL/min/1.73 m^2^ estimated by CKD-EPI equation	Prevalence of NAFLD was 100%. 47/217 subjects had CKD (21.6%).	PNPLA3 GG genotype was significantly associated with an increased risk of CKD in the whole population (aOR 3.42, 95% CI 1.07–10.85).
Liu HW et al., 2020 (Taiwan) [34]	Cross-sectional study	37,825 patients (13,288 men and 24,537 women)	Ultrasound	GFR < 60 mL/min/1.73 m^2^ estimated by CKD-EPI equation	Prevalence of NAFLD was 61.3% (23,209/37,825 patients). 4071/23,209 (17.5%) subjects developed incident CKD.	NAFLD was significantly associated with CKD (aOR 1.13, 95% CI 1.04–1.23). Prevalence of CKD increased with increasing severity of NAFLD (aOR 1.17, 95% CI 1.03–1.33).
Akahane T et al., 2020 (Japan) [32]	Cross-sectional study	3725 patients (1751 men and 1974 women)	Ultrasound	GFR < 60 mL/min/1.73 m^2^ estimated by MDRD Japanese equation	Prevalence of NAFLD was 31% (1154/3725 patients). 146/1154 (12.6%) subjects developed incident CKD. After propensity score matching, 138/1097 (12.6%) NAFLD patients developed CKD.	CKD prevalence was significantly higher in NAFLD patients only before propensity score matching. However, obesity (OR 2.10, 95% CI 1.40–3.17), hypertension (OR 1.50, 95% CI 1.02–2.22), and hyperuricemia (OR 2.41, 95% CI 1.54–3.79), were independent risk factors for CKD in patients with NAFLD.
An JN et al., 2020 (Korea) [37]	Prospective cohort study	455 NAFLD patients (221 men and 234 women)	Biopsy	GFR < 60 mL/min/1.73 m^2^ estimated by MDRD equation	Prevalence of NAFLD was 100%. 15/455 (3.3%) subjects developed incident CKD.	Risk of renal outcomes increased (aHR 5.63, 95% CI 1.81–17.51) with increased severity of portal inflammation. Patients with a higher score for portal inflammation (score 3–4) were found to be more prone to renal function deterioration, with a 7.7-fold increase in the risk of renal outcomes.
Chen PC et al., 2020 (Taiwan) [33]	Cross-sectional study	13,255 NAFLD patients (8710 men and 4545 women)	Ultrasound FLI and NFS to assess severity of fibrosis	GFR < 60 mL/min/1.73 m^2^ estimated by MDRD equation	Prevalence of NAFLD was 100%. 3195/13,255 (24.1%) developed CKD.	CKD patients had higher FLI and NFS than those without CKD. When subjects were stratified by NFS, CKD was found in 40.52% of patients with an NFS > 0.676. NFS > 0.676 was related to CKD with an OR of 2.266, 95% CI 1.560–3.291, and an aOR of 2.284, 95% CI 1.513–3.448.
Kaps L et al., 2020 (Germany) [41]	Retrospective cohort study	48,057 NAFLD patients (25,422 men and 22,635 women)	International Classification of Diseases, 10th revision (ICD-10)	International Classification of Diseases, 10th revision (ICD-10)	Prevalence of NAFLD was 100%. 8218/48,057 (17.1%) of patients were diagnosed with CKD.	Independent association of NAFLD with emerging CKD (HR 1.58, 95% CI 1.51–1.66).
Takahashi S et al., 2021 (Japan) [39]	Cross-sectional study	14,163 patients (9077 men and 5086 women)	FLI	GFR < 60 mL/min/1.73 m^2^ estimated by MDRD Japanese equation	2195/14,163 (15.5%) of patients developed CKD.	Higher FLI levels were independently associated with deterioration of renal function (aHR 1.33, 95% CI 1.16–1.54 in males; aHR 1.33, 95% CI 1.08–1.63 in females).

NAFLD, nonalcoholic fatty liver disease; CKD, chronic kidney disease; FLI, fatty liver index; GFR, glomerular filtration rate; CKD-EPI, Chronic Kidney Disease Epidemiology Collaboration; HR, hazard ratio; CI, confidence interval; NFS, NAFLD fibrosis score; APRI, AST-to-platelet ratio index; FIB-4, fibrosis-4 score; aHR, adjusted hazard ratio; MDRD, Modification of Diet in Renal Disease; aOR, adjusted odds ratio; OR, odds ratio; ICD-10, 10th revision of the International Classification of Diseases.

### 3.3. Main Findings

The main findings of the individual studies are reported in Table 1. In all but one study [32], an independent positive association between NAFLD and CKD was demonstrated. Akahane et al. observed a higher prevalence of CKD in NAFLD patients than in those without NAFLD before propensity score (PS)-matched analysis, but no significant difference between these groups in terms of CKD prevalence after PS matching. However, obesity, hypertension and hyperuricemia were demonstrated to be independent predictors of CKD in patients with NAFLD [32]. Furthermore, seven studies also found an association between CKD and the severity of NAFLD either assessed by noninvasive scores or by liver biopsy [30,31,33,34,35,37,39]. After NAFLD diagnosis, the severity of liver fibrosis was assessed by NFS in the majority of the studies. Nevertheless, the FIB-4 score was considered the best predictor of increased risk of CKD in the studies by Wijarnpreecha et al. [31] and Choi et al. [40]. Ӧnnerhag et al. diagnosed NAFLD by liver biopsy and assessed the severity of NAFLD also by noninvasive fibrosis scoring systems, reporting a moderately good predictive capacity of NFS and FIB-4 score for early identification of NAFLD patients at risk of CKD [35]. Furthermore, only one study reported an association between CKD and PNPLA3 GG genotype in patients with NAFLD [36]. Finally, two prospective studies demonstrated an increased incidence of CKD in patients with FLI-diagnosed steatosis [38] and in those with biopsy-proven NAFLD [37].

## 4. Discussion

The association between NAFLD and CKD has been of increasing interest in recent years. To date, accumulating evidence suggests that the pathogenesis of NAFLD is multifactorial. According to the multiple parallel hit hypothesis [42], hepatic insulin resistance, fatty liver infiltration, inflammation and chronic oxidative stress with the release of multiple proinflammatory cytokines play a major pathophysiological role in NAFLD development and progression to NASH. The same mechanisms may also contribute to the development and progression of CKD [43,44]. The crosstalk between the abnormal metabolic dysfunction and the systemic inflammatory responses leading to tubulointerstitial hypoxia may be considered the major mechanisms of CKD progression [45]. 

In 2014, the meta-analysis by Musso et al. (13 longitudinal studies with 28,963 participants) showed that the prevalence and the severity of NAFLD were associated with increased risk and severity of CKD [28]. In 2020, the last updated meta-analysis by Mantovani et al. [6], which incorporated 13 studies involving 1,222,032 individuals, confirmed the association between NAFLD and incident CKD. The authors suggested that the risk of CKD was higher among NAFLD patients with greater severity of liver fibrosis, but there were not sufficient data for a formal meta-analysis. Furthermore, none of the studies evaluated renal morphology and pathology associated with NAFLD. Therefore, it is currently still unknown if a specific type of kidney disease might be associated with NAFLD.

Our review study, including 12 articles with 180,010 participants, showed a strong association between NAFLD and CKD independently of traditional risk factors. Noteworthily, 7 out of 12 studies revealed the association between the increasing severity of NAFLD and the occurrence of CKD. Liu et al., in a large cross-sectional study, showed that the presence and severity of NAFLD, diagnosed based on abdominal ultrasonography, were significantly associated with CKD in adjusted analysis [34]. Sinn et al. also demonstrated an association between NAFLD and incident CKD which was stronger in participants with evidence of more advanced NAFLD, as indicated by a higher NFS score [30]. By contrast, Wijarnpreecha et al. reported that FIB-4 was found to be the most accurate predictor of CKD in U.S. adults with NAFLD [31]. These findings were confirmed by Chen et al. who reported in a large cross-sectional study that NAFLD per se was not a risk factor for CKD, but only patients with advanced liver fibrosis had a higher risk of CKD [33]. Similar results were obtained in two observational studies performed in biopsy-proven patients with NAFLD. In the first, noninvasive fibrosis scoring systems significantly predicted CKD [35], while in the study by Am et al., a higher score of portal inflammation was associated to a 7.7-fold increased risk of renal outcomes [37]. Finally, Takahashi et al. suggested the addition of FLI, a noninvasive marker of steatosis, into traditional risk factors for CKD because they found a discriminatory capability of high level of FLI for prediction of CKD in both sexes in a general population [39].

The mechanisms linking NAFLD, and CKD could be manifold. Both diseases share main risk factors, mostly represented by obesity and metabolic syndrome. Insulin resistance [19,46], systemic low-grade inflammation [47,48] and increased oxidative stress [49,50], induced by visceral obesity and metabolic dysregulation, can promote both NAFLD and CKD. Western diet, rich in added sugars and saturated fats, promotes insulin resistance and visceral obesity [51]. In addition, these nutrients exert a direct pathogenic role on the liver [52,53] and kidney [54]. Conversely, diets poor in saturated fats and added sugars and rich in vegetables and antioxidant nutrients (i.e., the Mediterranean diet) can protect from both NAFLD and CKD [55]. Finally, dietary habits can affect liver and kidney function, modulating gut microbiota and gut permeability [56,57]. 

Further evidence on the association between NAFLD and CKD, independent of metabolic diseases, may come from the reported association between PNPLA3 rs738409 G variant and CKD [36].

In fact, Sun D.Q. et al. found an association between PNPLA3 rs738409 G allele and renal glomerular and tubular injury in NAFLD patients with persistently normal ALT levels [36]. This finding is in accordance with that found in children by Targher et al. [58]. Interestingly, patients carrying this PNPLA3 gene variant do not have metabolic alterations usually shared by NAFLD and CKD. However, due to the small sample size, further analyses are needed. 

To the best of our knowledge, this is the last updated review on the association between NAFLD and its severity and the development of CKD. Our results showed that CKD developed more frequently in NAFLD patients compared to those without NAFLD. This association persisted after adjustment for traditional risk factors and according to the severity of NAFLD. Therefore, patients with NAFLD need to be carefully monitored for the development of CKD.

Our review may have some limitations and implications. A major limitation is that most of the studies considered are cross-sectional or retrospective. Moreover, the diagnosis of NAFLD was heterogeneous, ranging from liver ultrasound to biochemical variables (liver enzymes, FLI), even if liver biopsy is considered the gold-standard procedure. A strength of the study is the very large series of participants included in the review. Further strengths are that we included studies with sample sizes of no fewer than 100 patients and studies with no lack of detailed essential data. Finally, our review also suggests an increasingly important diagnostic role of several noninvasive hepatic steatosis and/or fibrosis tests such as FLI, NFS and FIB-4 score. Future clinical trials should focus on the specific role of these tests to improve the early diagnosis of CKD in patients with NAFLD and thus reduce the incidence of potentially life-threatening systemic complications.

## 5. Conclusions

In conclusion, the present study supports the hypothesis that NAFLD is an independent risk factor for the development of CKD probably because liver and kidney are closely and mutually intertwined. NAFLD may play a pathophysiological role in early CKD development and progression with NAFLD worsening to more severe stages.

Therefore, patients with NAFLD should be considered at high risk of CKD. Intensive multidisciplinary surveillance over time is needed, where hepatologists and nephrologists must act together for better and earlier treatment of NAFLD patients.

## Figures and Tables

**Figure 1 biomedicines-09-01370-f001:**
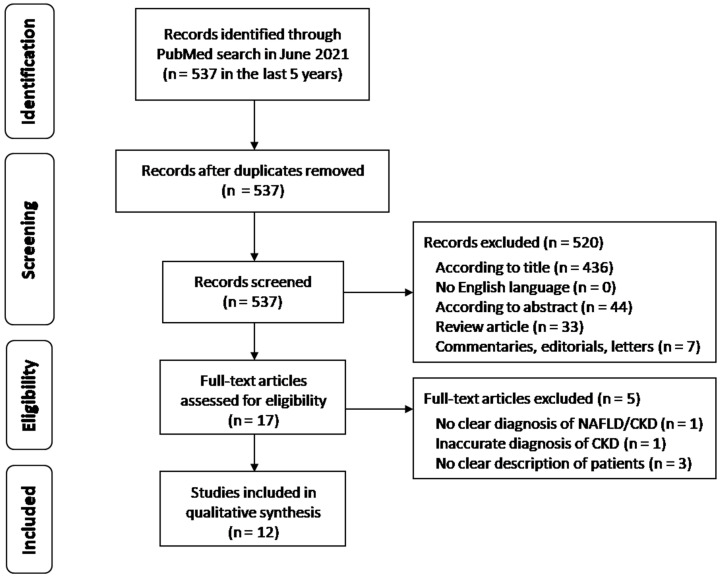
Flow chart of the search for relevant studies. NAFLD; nonalcoholic fatty liver disease, CKD; chronic kidney disease.

## Data Availability

Not applicable.

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
