# Peer review of "Nonalcoholic Fatty Liver Disease and the Kidney: A Review"

_biomedicines, 2021, doi:10.3390/biomedicines9101370_

Round 1
Reviewer 1 Report
The manuscript aimed to evaluate the correlation between NAFLD and the risk for chronic kidney disease.
The discussion section should be extended. More information regarding the pathophysiological mechanism involved in NAFLD-induced CKD. A schematic representation of the possible mechanisms will be also needed.
The strengths and limitations of the current review study should be added.
Author Response
September 23, 2021
Editorial Office
Biomedicines
MDPI
Dear Sirs
we would like to sincerely thank you and the Reviewers for your very useful comments which allowed us to significantly improve our work. We have prepared a point to point reply to the comments of Reviewers.
REVIEWER 1: The manuscript aimed to evaluate the correlation between NAFLD and the risk for chronic kidney disease. The discussion section should be extended. More information regarding the pathophysiological mechanism involved in NAFLD-induced CKD. A schematic representation of the possible mechanisms will be also needed. The strengths and limitations of the current review study should be added.
Reply: We thank the reviewer for your very useful comments which allowed us to significantly improve our work.
We have added the following paragraph in the “Discussion” section: “The mechanisms linking NAFLD, and CKD could be manifold. Both diseases share main risk factors, mostly represented by obesity and metabolic syndrome. Insulin resistance [44, 45], systemic low-grade inflammation [46, 47], increased oxidative stress [48, 49], induced by visceral obesity and metabolic dysregulation, can promote both NAFLD and CKD. Western diet, rich in added sugars and saturated fats, promotes insulin resistance and visceral obesity [50]. In addition, these nutrients exert a direct pathogenic role on liver [51, 52] and kidney [53]. Conversely, diets poor in saturated fats and added sugars and rich in vegetables and antioxidant nutrients (i.e., the Mediterranean diet) can protect from both NAFLD and CKD [54]. Finally, dietary habits can affect liver and kidney function modulating gut microbiota and gut permeability [55, 56]. Further evidence on the association between NAFLD and CKD, independent of metabolic diseases, may come from the reported association between PNPLA3 rs738409 G variant and CKD [34].”
Furthermore we have added limitations and strengths at the end of the Discussion section as follows: “Our review may have some limitations and implications A major limitation is that most of the studies considered are cross-sectional or retrospective. Moreover, the diagnosis of NAFLD was heterogeneous ranging from liver ultrasound to biochemical variables (liver enzymes, FLI) even if liver biopsy is considered the gold-standard procedure. Strength of the study is the very large series of participants included in the review. Further strengths are that we included studies with sample sizes no less than 100 patients and studies with no lack of detailed essential data. Finally, our review also suggests an increasingly important diagnostic role of several non-invasive hepatic steatosis and/or fibrosis tests such as FLI, NFS and FIB-4 score. Future clinical trials should focus on the specific role of these tests to improve early diagnosis of CKD in patients with NAFLD thus reducing the incidence of potentially life-threatening systemic complications.”
REVIEWER 2: In the article “Non-alcoholic fatty liver disease and the kidney: a review”, authors review the non-alcoholic fatty liver disease (NAFLD) association with chronic kidney disease (CKD). This review aims to provide a more comprehensive overview of this association, also considering the effect of increasing severity of NAFLD. The authors conduct a PubMed search using the terms “non-alcoholic fatty liver disease AND kidney”. In total, 537 articles were retrieved on PubMed search in the last five years nad finally, 12 articles were included in the qualitative analysis. This manuscript is timely, is well written and easy to follow The reviewer strongly recommends the authors address the comments below to improve this manuscript. Regarding the proposed renaming NALFD in MAFLD, please include the following articles in the introduction.
Baratta F, Ferro D, Pastori D, Colantoni A, Cocomello N, Coronati M, Angelico F, Del Ben M. Open Issues in the Transition from NAFLD to MAFLD: The Experience of the Plinio Study. Int J Environ Res Public Health. 2021 Aug 26;18(17):8993. doi: 10.3390/ijerph18178993. PMID: 34501590; PMCID: PMC8430687.
Wong VW, Wong GL, Woo J, Abrigo JM, Chan CK, Shu SS, Leung JK, Chim AM, Kong AP, Lui GC, Chan HL, Chu WC. Impact of the New Definition of Metabolic Associated Fatty Liver Disease on the Epidemiology of the Disease. Clin Gastroenterol Hepatol. 2020 Oct 31:S1542-3565(20)31504-4. doi: 10.1016/j.cgh.2020.10.046. Epub ahead of print. PMID: 33137486.
Reply: We thank the reviewer for the valuable and extremely positive suggestion. Following the Reviewer’s suggestion we have added the two references in the “Introduction” and “References” sections as number 17 and 18.
EDITORIAL OFFICE: After plagiarism check for your manuscript, we found some sentences share high similarity with published papers. File attached is the plagiarism report. Please rewrite the highlighted sentences with track-change (The similarity index should be less than 5%).
Reply: We thank the Editorial office for this useful comment which allowed us to significantly improve our work.
In the “Chronic kidney disease” section we replaced the paragraph “Furthermore, CKD was classified based on cause, GFR and albuminuria categories [25]. The cause of CKD has been traditionally assigned based on presence or absence of systemic diseases and location of observed or presumed pathologic-anatomic abnormalities. GFR categories are assigned as follows: G1, kidney damage with normal or increased GFR (≥ 90 mL/min/1.73 m2); G2, kidney damage with mildly decreased GFR (60-89 mL/min/1.73 m2); G3a, mildly to moderately decreased GFR (45-59 mL/min/1.73m2); G3b, moderately to severely decreased GFR (30-44 mL/min/1.73 m2); G4, severely decreased GFR (15-29 mL/min/1.73 m2); G5, kidney failure (<15 mL/min/1.73 m2). Importantly, the evidence of kidney damage is mandatory for the diagnosis of CKD in GFR categories G1 and G2 [25]. Albuminuria categories are assigned as follows: A1, normal, or mildly increased albumin-to-creatinine ratio (ACR < 30 mg/g); A2, moderately increased ACR (30-300 mg/g); A3, severely increased ACR (> 300 mg/g). Alternatively, reagent strips in spot urine can be used (A1, negative to trace; A2, trace to +; A3, + or greater) [25].” with the following “CKD can be classified on the basis of GFR categories as follows: G1, GFR ≥ 90 mL/min/1.73 m2; G2, GFR = 60-89 mL/min/1.73 m2; G3a, GFR = 45-59 mL/min/1.73m2; G3b, GFR = 30-44 mL/min/1.73 m2; G4, GFR = 15-29 mL/min/1.73 m2; G5, GFR < 15 mL/min/1.73 m2. Furthermore, CKD can be classified according to albuminuria categories as follows: A1, albumin-to-creatinine ratio (ACR) < 30 mg/g; A2, ACR = 30-300 mg/g; A3, ACR > 300 mg/g [25].”
In the “Materials and Methods” section we replaced the paragraph “Research was limited to title/abstract of articles published in English in the last 5 years; studies regarding animals and child, case reports, commentaries, letters, editorials, and meeting abstracts were not considered. Review articles were examined to identify studies that were potentially eligible for inclusion. Only potentially relevant studies underwent full-text review. Data were extracted on a standardized data collection table which included: first author, publication year, country, study design, number of patients involved, diagnosis of NAFLD, diagnosis of CKD, patient characteristics, results of the study.” with the following “Research was limited to studies published in English language over the last 5 years. Studies regarding animals and child, case reports, commentaries, letters, editorials, and meeting abstracts were not considered. Review-type articles were examined only to identify papers potentially eligible for inclusion. Only relevant studies potentially eligible for inclusion underwent full-text review. The following data were extracted on a data collection table: first author/publication year, study design, number of patients enrolled, diagnosis of NAFLD including assessment of severity of liver fibrosis, diagnosis of CKD, patient characteristics, main results of the study.”
In the “Results” section we replaced the paragraph “A flow chart of the search for relevant studies is presented in Figure 1. In total, 537 articles were retrieved on PubMed search in the last five years. After removal of irrelevant studies and exclusion according to title, language and abstract (n=520), 17 articles were selected for full-text review. A further 5 articles were excluded for the following reasons: 1 did not have a clear description of diagnosis of NAFLD and/or CKD, 1 had an inaccurate diagnosis of CKD and 3 did not have a clear description of patient’s enrollment and evaluation. Finally, 12 articles were included in the qualitative analysis.” with the following “A flow chart of the search for relevant studies is presented in Figure 1. Overall, 537 articles were found on PubMed search in the last 5 years. After exclusion of not significant studies according to title, language and abstract (n=520), 17 articles were selected for full-text review. A further 5 studies were excluded for the following reasons: 1 did not have a clear explanation of diagnosis of NAFLD and/or CKD, 1 had an inaccurate diagnosis of CKD and 3 did not clearly described patient’s enrollment and assessment. Finally, 12 papers were included in the qualitative analysis.”
I look forward to hearing from you at your earliest convenience.
Yours sincerely,
Ilaria Umbro, MD, PhD
ORCID: 0000-0001-8237-0766
Geramed Dialysis Center, Via Firenze 4,
00065 Fiano Romano, Rome, Italy.
Telephone number: 0765.455720
E-mail address: ilaria.umbro@geramed.it

Reviewer 2 Report
In the article “Non-alcoholic fatty liver disease and the kidney: a review”, authors review the non-alcoholic fatty liver disease (NAFLD) asociation with chronic kidney disease (CKD). This review aims to provide a more comprehensive overview of this association, also considering the effect of increasing severity of NAFLD.
The authors conduct a PubMed search using the terms”non-alcoholic fatty liver disease AND kidney”. In total, 537 articles were retrieved on PubMed search in the last five years nad finally, 12 articles were included in the qualitative analysis.
This manuscript is timely, is well written and easy to follow
The reviewer strongly recommends the authors address the comments below to improve this manuscript.
Regarding the proposed renaming NALFD in MAFLD, please include the following articles in the introduction.
Baratta F, Ferro D, Pastori D, Colantoni A, Cocomello N, Coronati M, Angelico F, Del Ben M. Open Issues in the Transition from NAFLD to MAFLD: The Experience of the Plinio Study. Int J Environ Res Public Health. 2021 Aug 26;18(17):8993. doi: 10.3390/ijerph18178993. PMID: 34501590; PMCID: PMC8430687.
Wong VW, Wong GL, Woo J, Abrigo JM, Chan CK, Shu SS, Leung JK, Chim AM, Kong AP, Lui GC, Chan HL, Chu WC. Impact of the New Definition of Metabolic Associated Fatty Liver Disease on the Epidemiology of the Disease. Clin Gastroenterol Hepatol. 2020 Oct 31:S1542-3565(20)31504-4. doi: 10.1016/j.cgh.2020.10.046. Epub ahead of print. PMID: 33137486.
Author Response

(The authors gave the same response as above.)

Round 2
Reviewer 1 Report
The authors addressed all my comments. The manuscript is much improved and ready for acceptance.